# Genomic Characterization and Expression Analysis of Basic Helix-Loop-Helix (bHLH) Family Genes in Traditional Chinese Herb *Dendrobium officinale*

**DOI:** 10.3390/plants9081044

**Published:** 2020-08-17

**Authors:** Yue Wang, Aizhong Liu

**Affiliations:** 1Key Laboratory of Economic Plants and Biotechnology, Yunnan Key Laboratory for Wild Plant Resources, Kunming Institute of Botany, Chinese Academy of Sciences, Kunming, 650201 Yunnan, China; wangyue@mail.kib.ac.cn; 2Bio-Innovation Center of DR PLANT, Kunming Institute of Botany, Chinese Academy of Sciences, Kunming, 650201 Yunnan, China; 3Key Laboratory for Forest Resources Conservation and Utilization in the Southwest Mountains of China, Ministry of Education, Southwest Forestry University, Kunming, 650224 Yunnan, China

**Keywords:** *Dendrobium officinale*, basic helix-loop-helix transcription factors, genome-wide identification, expression analysis, protein–protein interaction, skin care

## Abstract

*Dendrobium officinale* Kimura et Migo is of great importance as a traditional Chinese herb due to its abundant metabolites. The family of basic helix-loop-helix (bHLH) transcription factors widely exists in plants and plays an essential role in plant growth and development, secondary metabolism as well as responses to environmental changes. However, there is limited information on *bHLH* genes in *D. officinale*. In the present study, a total of 98 putative *DobHLH* genes were identified at the genomic level, which could be classified into 18 clades. Gene structures and conserved motifs in *DobHLH* genes showed high conservation during their evolution. The conserved amino acids and DNA bindings of DobHLH proteins were predicted, both of which are pivotal for their function. Furthermore, gene expression from eight tissues showed that some *DobHLH* genes were ubiquitously expressed while other *DobHLH* genes were expressed in the specific tissues. Expressional changes of *DobHLH* genes under MeJA and ABA treatments were detected by qRT-PCR. The protein–protein interactions between DobHLHs were predicted and several interactions were confirmed by yeast two hybrid. Therefore, our results here contribute to the understanding of *bHLH* genes in *D. officinale* and lay a foundation for the further functional study of its biological processes.

## 1. Introduction

*Dendrobium officinale* Kimura et Migo (or known as *D. catenatum*) is a perennial herbal plant, belonging to the family Orchidaceae. *D. officinale* plants are widely distributed across southern provinces in China, with limited spread to southeast Asian countries [1]. *D. officinale* is of great importance due to its medicinal and ornamental uses. *D. officinale* ranks first among the nine Chinese herbs for longevity and has a documented use in folk medicine for over 1300 years [2]. In the past decades, about 190 compounds have been identified and isolated from *D. officinale* plants including phenanthrenes, bibenzyls, saccharides and glycosides, alkaloids, essential oils, and others such as phenols, acids, esters, and amides [3]. Among them, polysaccharides, bibenzyls, alkaloids, and flavones have been confirmed as the major and bioactive ingredients [3,4]. The abundance of polysaccharides in its stems and the specific alkaloids make *D. officinale* have anticancer [1], anti-oxidant [5], and anti-inflammatory effects [1,6] as well as support immune modulation [7,8], hepatop protection, and hypoglycemia [8,9]. Due to its multiple health benefits, demand for high-quality *D. officinale* is increasing across medicine, food, and skin care.

As an epiphytic plant, wild *D. officinale* prefers to grow in a warm and humid environment. In nature, *D. officinale* inevitably suffers environmental stresses such as high temperature and drought, slowing growth, and lowering yield [10]. Despite abundant polysaccharides, the contents of other active components in *D. officinale* are low [11,12,13]. At present, many potential genes involved in the biosynthesis of polysaccharides, alkaloids, and flavonoids have been identified in *D. officinale* through transcriptome and genomic sequencings [4,14]. In *D. officinale*, polysaccharides in the stem, leaves, and flowers are one of the major active ingredients, and sucrose synthase (Susy) and two gene families (β-galactosidase and galacturonosyltransferase) are related to the contents and richness, respectively [2]. In addition, 10 sucrose phosphate synthase and 15 Susy genes showed marked expansion [2]. The upstream pathways of alkaloid biosynthesis in plants are conserved and clear, which are responsible for the formation of strictosidine. The downstream steps of the alkaloid biosynthesis in *D. officinale* are modification of strictosidine including cytochrome P450s (CYP450s)-mediated oxidation and hydroxylation reactions [15], which needs further investigation. However, transcription regulation of the biosynthesis has been largely unknown. As an important switch of gene expression, transcription factor (TF) can activate or repress the expression of specific target genes through interaction with cis-elements in the gene promoter region, which moderates various biological processes such as growth, response to stresses, and the biosynthesis of secondary metabolites [16].

Basic helix-loop-helix (bHLH) TFs are widespread in the eukaryotes and also are the second largest family of TFs in plants [17,18,19,20]. bHLH TFs are named due to the presence of a highly conserved bHLH domain, and the bHLH domain of 50–60 amino acids contains one basic region and one HLH region [21,22]. The basic region of 15 amino acids is at the N-terminus of the bHLH domain, while the HLH region of 40–50 amino acids is at the C-terminus of the bHLH domain including two alpha helices and one less conserved loop with variable lengths. The bHLH domain can specifically recognize and bind a conserved cis-element CANNTG, named E-box [23,24]. The two helices from the same bHLH TF or different bHLH TFs can interact to form a homodimer or a heterodimer, which then binds different regions in the promoter to regulate the expression of their target genes [25]. The other sequences of bHLH TFs are variable and less conserved.

The bHLH TF was first found in murine muscle [21], and later in animals [23] and plants [22,26]. Based on differences in the sequences, DNA binding, bHLH domain and functions, the bHLH TF family in animals is divided into six lineages (group A–F) [23]. In plants, most bHLH TFs share a similar structure with that of group B from animals, which can bind the G-box (CACGTG), although they also bind E-box [16]. Unlike in animals, Arabidopsis bHLH TFs are divided into many more clades [22]. The classification of plant bHLHs is gradually being clarified and most plants contain 15–26 clades due to their identification in more plants including tomato [27], maize [28], grape [29], cotton [30], jujube [25], and *Brachypodium distachyon* [31].

bHLH TFs play important roles in plant secondary metabolites. The first bHLH protein identified in plant was *R* gene in maize, which can regulate the expression of at least two genes in the flavonoid/anthocyanin pathway [32]. In Arabidopsis, the three bHLH proteins Glabra3 (GL3), Enhancer of Glabra3 (EGL3), and Transparent Testa8 (TT8) can interact with the myeloblastosis (MYB) transcription factor and WD40 protein to form a MYB–bHLH–WD complex, activating multiple genes in the biosynthesis of anthocyanin and resulting in anthocyanin accumulation [33]. An increasing number of research is reporting that bHLH TFs also participate in the regulation of other secondary metabolites including flavonols [33], terpenoid indole alkaloids [34], and tanshinone [35]. Recently a bHLH TF was found to be responsible for the specific anthocyanin of lip tissues in *Dendrobium* hybrids [36]. Identification of bHLH TFs in *D. officinale* is the first step to understanding their roles in the accumulation of bioactive compounds.

In addition, bHLH TFs participate in plant growth and development including photomorphogenesis, light signal transduction, flowering time, and the development of various tissues such as flower and root [16]. For example, two Arabidopsis phytochrome interacting factors (AtPIF4 and PIF3) can interact with phytochrome to control the expression of the genes involved in the regulation of light response [22,37]. A large number of bHLH TFs play a vital role in the response to environmental stresses such as salt, drought, and high temperature. Inducer of CBF expression 1 (ICE1), belonging to the III b subclade of Arabidopsis bHLH family, can activate downstream genes through the mediation of C-repeat binding factor (CBF) in response to freezing stress [38].

Although there has been much research on bHLH TFs in various plants, a comprehensive investigation of the bHLH family in *D. officinale* has yet not been carried out. The competence of genome sequencing for *D. officinale* makes the search for potential genes associated with important traits possible [2,39]. In the present study, we first identified a total of 98 candidate DobHLH members. Their phylogenetic relationship, conserved bHLH domain and motifs, gene structures, and cis-elements were characterized. The expression patterns in eight tissues were analyzed from the RNA-Seq data. The expression levels of some *DobHLH* genes under hormone treatments were detected by quantitative real time-polymerase chain reaction (qRT-PCR). The possible protein–protein interactions were predicted, and some interactions were confirmed by yeast two hybrid. Accordingly, our results provide information on the bHLH family in *D. officinale* and lay the foundation for further investigation into its function in its biological process.

## 2. Results

### 2.1. Genome-Wide Identification of DobHLH Members in Dendrobium officinale

After the hidden markov model (HMMER) search and confirmation of the bHLH domain presences with online CD-search tool and SMART, a total of 98 members in *D. officinale* were considered DobHLH candidates. They were named from DobHLH1 to DobHLH98 according to their relationship with AtbHLH proteins. The basic information of DobHLH members is shown in Table 1 and Appendix A. The DobHLH proteins contain 85–662 AAs with molecular weights of 9.53–74.54 kDa and isoelectric points of 4.41–10.78. The subcellular localization of 98 DobHLH proteins were predicted by WOLF PSORT. The results showed that most of the predicted proteins (89) were localized in the nucleus, and five and four DobHLH proteins were in the cytoplasmic and mitochondrial, respectively. The 98 DobHLH genes were distributed in 85 scaffolds. The CDS lengths of the *DobHLH* genes varied from 258 bp (*DobHLH95* and *DobHLH96*) to 1989 bp (*DobHLH24*) while the lengths of genomic DNA were from 496 bp (*DobHLH79*) to 26,581 bp (*DobHLH58*).

Compared with the numbers of bHLH members in other plant species, the number in *D. officinale* was similar to that found in grape [29] and Chinese jujube [25] and was less than that found in most other species with over 100 bHLH members such as Arabidopsis (166), rice [40], *Brachypodium distachyon* [31], and tomato [27]. Some important crops had up to 200 or even many more bHLH members including maize [28], poplar [41], wheat [42], cotton [30], and *Brassica napus* [43]. The density of bHLH members in the *D. officinale* genome was approximately 0.28, which was similar to peach, but higher than that found in the genomes of the two lower plants *Volvox carteri* (0.024) and *Chlorella vulgaris* (0.081) [44]. These results showed that the number of *bHLH* genes in different species could be connected with the genome size.

### 2.2. Phylogenetic Relationship of DobHLH Proteins

To reveal the evolution of bHLH members, an unrooted-tree was constructed using MEGA 7.0 with the neighbor-joining method including 98 DobHLHs and 166 AtbHLHs (Figure 1). The phylogenetic tree could be divided into 22 clades, and each clade included two to 31 members. Ninety-eight DobHLHs were distributed in 18 clades (clade 1 to clade 18) and four clades (clade 19, 20, 21, 22) contained bHLH proteins only from Arabidopsis, without proteins from *D. officinale*, which suggested that gene deletion could occur during the evolution of *D. officinale*. In most clades, there were fewer bHLH proteins from *D. officinale* than from Arabidopsis. Clade 4 and 6 had more DobHLHs than AtbHLHs and clade 9 and 16 had the same number of bHLH members (three and one).

Moreover, an unrooted tree was constructed with bHLH proteins from *D. officinale* and rice (Appendix A). For as many as 15 clades, there were fewer bHLH proteins from *D. officinale* than from rice. Clades 6 and 17 contained more bHLH proteins from *D. officinale* than from rice. There were more bHLH proteins in the rice genome than in *D. officinale*, which could result from genomic or random duplication after differentiation with *D. officinale* [40].

### 2.3. Gene Structures and Conserved Motifs of DobHLH Genes

To reveal the evolution of *DobHLH* genes, the exon-intron structures were investigated by comparing CDS sequences with genomic sequences. The results showed that the intron numbers varied greatly among the DobHLH family members, from some completely lacking intron to some having up to 11 introns (Figure 2). Most of the members within the same clade shared the same number of introns. For example, clades 1, 2, 8, 10, 12, 16, 17 had two, three, one, six, five, two, and one intron, respectively, while all members in clade 6 and eight members in clade 15 had no introns. Appropriately 25% of the members (25) had one or two introns, mainly including members from clades 1, 5, 8, 16, 17 and 18. Most members from clades 3, 5, 7, 10, 11, 12, 13, and 14 including 43 DobHLH genes contained three to seven introns. Four genes (*DobHLH24*, *42*, *43*, and *64*) had more than seven introns. The lengths of introns imbedded in the *DobHLH* genes ranged from 44 bp to 21,971 bp, with the longest intron in the *DobHLH53* gene.

To further understand the structure of DobHLH proteins, we searched 10 conserved motifs in DobHLHs with MEME software. As shown in Figure 3, motif 1 and motif 2 were the two most conserved motifs, which were widely present in DobHLH members. The latter ten residues in the basic region and the first helix were motif 1, while the second helix was the main part of motif 2. Motif 1 was present in 88 DobHLH proteins while motif 2 was present in all the 98 DobHLH proteins. Ten DobHLH proteins (including DobHLH13, 14, 16, 17, 43, 92, 94, 95, 96, 98) had motif 2 only and lacked the basic region of bHLH domain. The other 88 DobHLH proteins had both motif 1 and motif 2, and 30 DobHLH members contained only motif 1 and motif 2, mainly from clades 4, 7, 8, 9, 11, 15, 16, and 17. Clades 5 and 6 contained the most motifs (five to seven). Motif 3 was present in all the members from clades 12, 13, 14 and some members from clade 15. Motif 3 was distributed in clade 1, 2, 5, and 6, with the exception of DobHLH25 and DobHLH26. Motif 5 and motif 6 were both present in the genes of clade 5 and clade 6. Motif 7 was limited in two members from clade 5 (DobHLH22 and DobHLH23) and seven members from clade 6. Motif 8 was present in all genes from clade 10 and clade 13. Motif 9 was limitedly present in the members of clade 6 without DobHLH29. Motif 10 was present in nine members from clades 1 and 6 (DobHLH1-6, 8, 9, 25, 26). There were similar motifs in most members from each clade. Eight of nine DobHLH members from clade 1 contained four motifs (motif 1, 2, 4, 10), and three members from clade 2 had motif 1, 2, and 4. Four of five members in clade 3 included motif 2. Ten DobHLH members from clade 10 had motifs 1, 2, and 8 while 14 members from clade 14 included motifs 1, 2, and 3. 

### 2.4. Conserved Amino Acids in the DobHLH Domains and DNA-Binding Ability

As the bHLH domain composed of motif 1 and motif 2 is especially important for bHLH TF to function, we analyzed the amino acids in bHLH domains from 98 DobHLH proteins (Figure 4). It was found that the *D. officinale* bHLH domain included 55 AAs, which was similar in Arabidopsis (56, [22]) but shorter than in tomato (61, [27]). The basic region in the *D. officinale* bHLH domain included 17 AAs and five AAs, showing over 50% identity in sequences. The first helix and the second helix both contained 15 AAs and six and eight AAs had more than 50% identity, respectively. The loop in the *D. officinale* bHLH domain had AAs and only Asp-40 was conserved with an identity of 60%.

As seen in Appendix A, there were 20 AAs in the *D. officinale* bHLH domain that showed over 50% conservation and three residues (Arg-16, Leu-27, and Leu-55) were highly conserved in *D. officinale*, up to 89%, 96% and 91%, respectively. It has been found that these three residues were conserved both in plants and animals [22], suggesting their core sites in the bHLH domain. Moreover, seven residues including Ile-20, Leu-24, Gln-28, Met-44, Ile-49, Val-52, and Leu-55 were more conserved in plants such as *D. officinale* and Arabidopsis than in animals, which was also observed in eggplant [22]. This could imply their important role in plants. Compared with residues in tomato and Arabidopsis, Glu-13, Arg-14, Gln-28, Lys-36, Ala-48, and Ile-52 were less conserved. 

As TFs, the most important function for bHLH proteins is to bind the promoter region of their target genes. Based on previous studies on the ability to bind DNA, we distinguished the types of DobHLH proteins (Table 2). It was found that 98 DobHLH proteins could be divided into two types: 22 non-DNA binding DobHLHs and 76 DNA binding DobHLHs. The 22 non-DNA binding DobHLHs mainly included members from clades 2, 10, 12, 15, and 17, and there were less than six residues in the basic region of 17 AAs in the bHLH domain. According to differences in the DNA binding sequences, 76 DNA binding DobHLHs could be further classified into 65 E-box binders and 11 non-E-box binders. When the 13 and 16 residues in the basic region are Glu and Arg, respectively, the bHLH protein has the capacity to bind E-box. Otherwise, the DobHLH protein belongs to non-E-box binders. Eleven non-E-box binders were from clades 10, 15, 17, and 18. The E-box binders could be further discriminated into G-box binders and non-G-box binders. When the 9, 13, and 17 residues in the basic region are His/Lys, Glu and Arg, respectively, the bHLH protein is considered to bind the classic G-box. In *D. officinale*, there were four DobHLH proteins that belong to non-G-box binders including DobHLH34, 35, and 36 from clade 7 and DobHLH97 from clade 17. The remaining 61 DobHLH proteins were G-box binders, accounting for 62.24% of all the DobHLH members.

### 2.5. Cis-Elements in the Promoter Regions of DobHLH Genes

Due to functional divergence, the members in the same gene family could show diverse expression patterns. To investigate regulatory patterns, we detected the cis-elements in the promoter regions of *DobHLH* genes (Appendix A). Many cis-elements related to responses to stresses were predicted by PlantCARE. The most frequent cis-element was G-box, which was present in 72 *DobHLH* genes, suggesting that a light signal could be vital in transcriptionary regulation. Other cis-elements such as the CGTCA-motif (or TGACG-motif), ABRE, and ERE were distributed in 65, 69, and 78 *DobHLH* genes, respectively, which are responsible for responses to methyl jasmonate (MeJA), abscisic acid (ABA), and ethylene, respectively. There were 166 CGTCA-motifs (or TGACG-motifs), 201 ABRE cis-elements, and 188 ERE cis-elements. The cis-elements related to stress responses were also found in *DobHLH* gene promoter regions such as TC-rich repeats in 47 *DobHLH* genes, WUN-motif in 50 DobHLH genes, LTR in 39 *DobHLH* genes, and MBS in 42 *DobHLH* genes. These cis-elements can respond to defense and stress, wounding, low temperature, and drought. Other cis-elements involved in plant growth and development were observed in some DobHLH genes including ARE, CAT-box, and O2-Site.

### 2.6. Expression Patterns of DobHLH Genes in Eight Tissues

The expression patterns of *DobHLH* genes were analyzed based on the transcriptomic data of eight tissues including column, flower buds, lip, sepal, leaf, stem, white root, and green root tip. There were 23 *DobHLH* genes that were hardly detected in any of the eight tissues The expression of 23 *DobHLH* genes was not detected in any of the eight tissues (when FPKM values were < 2, the gene was considered to be not expressed). Those 23 *DobHLH* genes were members from clades 1, 4, 10, 14, and 15. The other 75 *DobHLH* genes were expressed in at least one tissue tested with FPKM > 2. A heatmap of the expression patterns including 75 *DobHLH* genes was constructed by R, as shown in Figure 5. The genes could be classified into five groups. 

Group I included six genes (*DobHLH7*, *DobHLH81*, *DobHLH83*, *DobHLH87*, *DobHLH94*, and *DobHLH96*). Their transcripts could not be detected across four vegetative tissues (leaf, stem, white root, and green root tip) and were expressed with various levels in four flower tissues. *DobHLH81* and *DobHLH83* were lowly expressed only in the column. *DobHLH87* was moderately expressed in flower buds while the transcripts of *DobHLH96* were abundant in flower buds with FPKM of 124.68. *DobHLH7* and *DobHLH87* were expressed in four flower tissues with different levels.

Group II included 13 genes (*DobHLH1*, *2*, *4*, *10*, *20*, *24*, *35*, *38*, *47*, *72*, *76*, *77*, and *79*). Only two genes (*DobHLH10* and *DobHLH77*) were moderately expressed in certain tissues (white root and flower buds, respectively). The other genes were not expressed in most tissues and expressed only in one or two tissues with low FPKM values (<7.50).

Group III included four genes (*DobHLH11*, *DobHLH12*, *DobHLH92*, *DobHLH93*) that showed a strong tissue-specific expression pattern. These four genes were low expressed in root tissues. *DobHLH11* and *DobHLH12* were expressed in both white root and green root tip. *DobHLH12* was highly expressed in both white root and green root tip, while *DobHLH11* was highly expressed in the white root, but low in the green root tip. The transcripts of *DobHLH92* and *DobHLH93* were limitedly detected in the white root, but not the green root tip. *DobHLH93* showed the highest expression level in the white root with FPKM of up to 286.63.

Group IV contained 40 *DobHLH* genes. According to their expression differences, these genes could be further divided into five subgroups. Subgroup 1 included 11 genes (*DobHLH18*, *27*, *28*, *30*, *37*, *40*, *49*, *50*, *59*, *66*, *68*, and *97*). *DobHLH30*, *37*, and *66* were not expressed or lowly expressed in most tissues and highly expressed in some tissues. For example, *DobHLH30* showed the highest expression with FPKM of 231.28 in the column, followed by FPKM of 61.66 in the sepal. *DobHLH37* was the most abundant with FPKM of 42.56 in the stem while *DobHLH66* was expressed with the highest FPKM of 40.61 in the white root. The remaining eight *DobHLH* genes were not expressed or expressed with low FPKM values. Subgroup 2 contained five members (*DobHLH22*, *DobHLH26*, *DobHLH28*, *DobHLH33*, and *DobHLH69*). With the exception of *DobHLH28*, which was moderately expressed in the lip, all these genes were not expressed or lowly expressed in the leaf, lowly expressed in the stem and lip (5 < FPKM < 10), moderately expressed in the white root and green root tip, and moderately or abundantly expressed in the column, flower buds, and sepal. Subgroup 3 contained 12 genes (*DobHLH31*, *32*, *41*, *42*, *55*, *56*, *57*, *58*, *60*, *62*, *63*,and *65*) that were widely expressed in all tested tissues with various levels. All these genes were moderate or abundant with FPKM values of more than 14.68 in the green root tip. For six genes (*DobHLH32*, *42*, *57*, *60*, *63*, and *65*), there were more transcripts in the white root and green root tip than in other tissues tested. Except for *DobHLH42*, other genes had higher expression levels in the green root tip than in the white root. *DobHLH31*, *41*, *55*, *56*, *58*, *62*, and *63* genes showed the highest expression levels in non-root tissues. Subgroup 4 contained seven genes (*DobHLH3*, *17*, *25*, *43*, *48*, *52*, and *53*). Most of them were ubiquitously expressed in eight tissues tested with low or moderate levels. *DobHLH52* showed the highest expression level with FPKM of 35.73 in the leaf. Subgroup 5 contained five genes (*DobHLH5*, *34*, *36*, *70*, and *95*). These genes showed no or low expression levels in the lip, leaf, stem, white root, and green root tip, exhibiting moderate levels in the column, flower buds, and sepal tissues. *DobHLH95* showed the highest level with FPKM of 64.94 in the sepal and *DobHLH34* contained a maximum FPKM of 63.41 in the column.

Group V contained 12 genes (*DobHLH13*, *14*, *15*, *16*, *23*, *39*, *54*, *64*, *73*, *74*, *75*, and *98*). Only *DobHLH98* was expressed in the green root tip with a low level. All other genes were moderately or highly expressed in all tested tissues with FPKM of more than 11.89, suggesting their various roles in the eight tissues. *DobHLH13* and *DobHLH16* had the most abundant transcripts in the green root tip with FPKM values of 122.75 and 140.77, respectively. *DobHLH73* was highly expressed in the column and flower buds. DobHLH54 had the most transcripts in the leaf with FPKM of 151.78, while *DobHLH39* showed the highest expression level in the stem with FPKM of 80.19.

To further confirm the expression profiles of *DobHLH* obtained from RNA-Seq data, eight genes (including *DobHLH11-16*, *23,* and *52*) were randomly selected for the performance of qRT-PCR assays. The results showed that most of the genes exhibited similar expression patterns to RNA-Seq data (Appendix A). For example, *DobHLH11* and *DobHLH12* genes were expressed highest in roots and *DobHLH13*, *14*, *15,* and *16* were expressed in the root, stem, and leaf tissues, consistent with RNA-Seq data. However, the expression pattern of *DobHLH23* showed a slight difference from the RNA-Seq data.

### 2.7. Expressional Changes of DobHLH Genes Concerned with ABA and JA Signals under MeJA and ABA Treatments

To investigate whether *DobHLH* genes that contain stress-related regulatory cis-elements could respond to different stresses, we simulated the stress treatment experiments using exogenous MeJA and ABA treatments and inspected the expressional changes of 16 *DobHLH* genes concerned with ABA and JA signals by the qRT-PCR technique. As shown in Figure 6, eight (including *DobHLH* 3, 10, 15, 16, 23, 31, 33, and 37) of 12 *DobHLH* genes that contained JA-responsive elements (CGTCA-motif cis-element) exhibited expressional changes. In contrast to the controls, four *DobHLH* genes (including *DobHLH12*, *13*, *14* and *32*) exhibited no expressional changes in response to MeJA treatments. Similarly, with ABA treatment, 11 (including *DobHLH* 3, 11, 12, 15, 16, 23, 30, 32, 33, 37, and 39) of 13 *DobHLH* genes that contained ABA-responsive elements (ABRE-motif cis-element) exhibited expressional changes compared to the controls, whereas two *DobHLH* genes (*DobHLH13*, *14*) exhibited no expressional changes in response to ABA treatments. In particular, we noted that six *DobHLH* genes (*DobHLH* 3, 15, 16, 23, 33, and 37) of the ten *DobHLH* genes (including *DobHLH* 3, 12, 13, 14, 15, 16, 23, 32, 33, and 37) contained both JA and ABA-responsive elements, which showed expressional responses to both ABA and MeJA treatments. However, *DobHLH13* and *14* did not respond to both ABA and MeJA treatments at the transcription level. These results suggest that most of the *DobHLH* genes might function in respond to stress signals at the transcription level, likely through specific cis-elements in promoter regions, though the processes of regulatory transcription might be complex.

### 2.8. Protein–Protein Interaction Networks

It has been reported than bHLH proteins can form a homodimer or a heterodimer to bind DNA sequences in the promoter region to regulate the transcription of the target gene. Therefore, interactions between bHLH proteins play an important role in their function. The predicted protein–protein interaction of DobHLH proteins based on homologs with Arabidopsis bHLH proteins is shown in Figure 7. It was observed than most DobHLH proteins could interact with one or more DobHLH proteins. For example, it was predicted that DobHLH17 can interact with multiple DobHLH proteins including DobHLH6, 8, 9, 13, 14, 15, 16, and 60. The homolog of DobHLH17 in Arabidopsis is POPEYE (PYE), which is involved in the regulation of responses to iron deficiency in Arabidopsis roots. It has been demonstrated that PYE can interact with multiple proteins such as UNE12, AtbHLH104, ILR3, FMA, MUTE, and SPCH. ILR3 plays a role in the development of root hair. The interaction between DobHLH17 and DobHLH13 or DobHLH14 could have a similar function in *D. officinale*. Moreover, FMA, SPCH, and MUTE together regulated the stomata formation in Arabidopsis and ICE1 could interact with FMA, SPCH, and MUTE [25], implying that their homologs in *D. officinale* could potentially participate in the regulation of stomatal differentiation. HEC2 and SPT both play a role in floral development [25], and interactions between their homologs in *D. officinale* (DobHLH77, 78, 79, and DobHLH53) could influence pistil development through the control of hormones. These predicted interactions suggest that DobHLH proteins could function together to regulate plant growth, development, and other biological processes that merit further investigation.

To further verify protein–protein interactions in the predicted network, we randomly selected 12 *DobHLH* genes to investigate their interactions using the Y2H technique in yeast. Based on their co-expressions in a given tissue, we made different combinations in Y2H experiments. The selected *DobHLH* genes were cloned into AD and BD vectors, respectively. As shown in Figure 8, none of the DobHLH proteins could interact with the empty BD vector, while three DobHLH proteins (DobHLH36, 48, and 70) could interact with the empty AD on SD/-Leu-Trp-His-Ade medium, suggesting their self-activation. In the combination of DobHLH proteins, it was found that DobHLH25 and DobHLH26 showed an interaction on the SD/-Leu-Trp-His medium and DobHLH60 and DobHLH17 showed a strong interaction on the SD/-Leu-Trp-His-Ade medium. Other protein–protein combinations had no interaction. These results can partially confirm the predicted interaction networks and indicate that they could play similar functions in *D. officinale*.

## 3. Discussion

TFs can bind cis-element sequences in downstream genes to activate or repress their expression, which confers vital roles in plant growth, development, and responses to stresses [16]. The genome sequences of *D. officinale* have been reported [2,39], which makes it possible to identify the TF family at the genome level. The bHLH family is the second largest family of TFs and plays an essential role in plants. In the present study, based on the published genome data, bHLH family members were first identified in *D. officinale*, which provides valuable information to dissect their potential function in molecular and physiological processes.

A total of 98 DobHLH members were identified according to the genome data [39]. In previous transcriptome data, some bHLH members were predicted [13,14,15,39,45] and the number of bHLH members varied amongst these publications. The number of bHLH members identified by the genome in the present study was greater than the numbers by the transcriptome data. This was because some bHLH genes were limitedly expressed in the specific tissues and could not be detected in the specific tissues used for RNA-Seq. Only in three transcriptomes from the root, stem, and leaf were there as many as 140 *bHLH* transcripts, which could result from alternative splicing events [13]. Compared with other TF families identified in *D. officinale*, the number of bHLH family members ranked second, after the MYB family [46], and was greater than other families such as GRAS [10] and MADS [47]. The number of bHLH members in *D. officinale* was similar to that in jujube [25] and peach [44] and was more than in *Carthamus tinctorius* [48]. It has been found that many crops contain more than 150 bHLH members including maize [28], wheat [42,49], and *Panax ginseng* [50].

The bHLH members in plants could be divided into different clades (or subfamilies) and the numbers of clades in different species varied from 15 to 25 [22]. In the present study, the 98 DobHLH genes in *D. officinale* could be classified into 18 clades. We found that four clades in Arabidopsis were not present in *D. officinale*. The similar results were found in other plants such as *C. tinctorius* [48], apple [51], and wheat [42], which suggests those genes were lost during their evolution. Clades 14 and 15 of the bHLH family in *D. officinale* contained 14 and 15 members, respectively, which accounted for as many as 28.57% members of the family. The expansion of two clades in *D. officinale* could potentially imply their important functions. However, the numbers of members from clade 7 and 11 in *D. officinale* were lower than those in Arabidopsis and rice, which suggests that these clades shrunk in *D. officinale*.

It was found that the bHLH family in *D. officinale* showed conservation in both gene structure and protein motif. The gene structure of bHLH family in *D. officinale* showed highly conserved patterns. Most members in the same clade had the same number of introns and all numbers from clade 6 had no intron. Similar results were observed in *Nelumbo nucifera* [52], apple [51], cotton [30] and rice [53], which further suggested conserved gene organization during the evolution of different species. However, there were some exceptions where some members in the same clade contained variable intron numbers such as clade 14, which had five, six, seven, or 10 introns. The events of exon loss and gain occurred during the evolution of these genes, which could result in functional diversities between closely-related genes.

The most conserved motifs in the DobHLH proteins were motif 1 and motif 2, which consisted of the bHLH domain. As a core domain for the bHLH family, the bHLH domain was also highly conserved in other species such as tomato [27], wheat [42], and maize [28]. In the bHLH domain of *D. officinale*, the most conservative residues were Leu-27 and Leu-54, which were in consistence with the residues in Arabidopsis, tomato, and rice [22,27,53]. These results suggest that the two residues may be essential for their functions, especially for the formation of dimers [27]. Moreover, most residues in the domain had similar conservation as in Arabidopsis and tomato, while some residues in *D. officinale* showed less conservation than those in Arabidopsis and tomato including Glu-13, Arg-14, Gln-28, Lys-36, Ala-48, and Ile-52.

As TFs, their most important function is to bind the specific DNA sequences to regulate the gene expression. According to previous reports and the bHLH domain, the potential types of DNA-binding for DobHLHs were predicted in *D. officinale*. Compared with predicted DNA-binding types in Arabidopsis and tomato, the number of DNA bindings (77.55%) in *D. officinale* was lower than that in Arabidopsis (81.63%), but higher than in tomato (69.18%). Among them, the number of G-box DNA binding bHLHs (61/98, 62.24%) in *D. officinale* was similar to that in Arabidopsis (89/147, 60.54%), which was much higher than that in tomato (72/159, 45.28%). The number of the non-E-box binders in *D. officinale* was slightly higher than that in Arabidopsis as well as in tomato. The comparative results suggest that a large number of bHLH proteins in *D. officinale* could bind DNA similar to other species. Notably, most members from the same clade showed the same type of DNA-binding, which was also observed in other species such as Arabidopsis and tomato [27]. Furthermore, the bHLH proteins from *D. officinale* and Arabidopsis in the same clade exhibited the same type of DNA-binding, which confirmed their conservation during evolution.

The cis-regulatory elements are located at the 5′ upstream of the gene and are responsible for gene transcription, which plays an essential role in the regulatory network and control of plant growth and development. Many cis-elements related to the development, responses to stress as well as hormone were predicted in the promoter regions of *DobHLH* genes, which suggest their potential roles in the responding conditions. The most common cis-element observed in *DobHLH* gene promoters was the G-box, which can respond to light. Similar results were found in other species such as *C. tinctorius* [48] and maize [28]. The second most common types of cis-elements in DobHLH genes were the hormone-responsive motifs that responded to MeJA, ABA, and ethylene. Among 98 *DobHLH* genes, there were as many as 201 ABRE elements in 69 *DobHLH* genes, suggesting their putative roles in response to ABA. Notably, a total of 188 ethylene-responsive ERE elements were predicted in 78 *DobHLH* genes, while few ERE elements were observed in *bHLH* genes from other plants. The rest of the cis-elements including TC-rich repeats, LTR element, and MBS were also predicted in *DobHLH* gene promoters, which were related to abiotic stresses such as low temperature and drought. The results showed that *bHLH* genes can respond to various environmental changes through their cis-elements.

There has almost been no report about bHLH functions in *D. officinale* despite many reports on other plants such as Arabidopsis and rice. The potential gene function could be predicted based on its expression pattern. In the present study, the comprehensive expression patterns of *DobHLH* genes were detected in eight tissues according to the previous RNA-Seq data, which strongly showed tissue-specific expression. A total of 23 *DobHLH* genes were barely expressed in these tissues. Similarly, some *bHLH* genes from other species were also not detected in any tissue [49]. For example, there were 30 *bHLH* genes in potato that were not expressed in any of the 12 tested tissues [54]. However, some *bHLH* genes could widely be expressed in various tissues such as the 12 *bHLH* genes in *D. officinale* including *DobHLH13* and *DobHLH16*. These ubiquitous and abundant transcripts of *bHLH* genes could be found in jujube [25] and potato [54]. The widespread expression of *bHLH* genes in multiple tissues suggests their multiple roles in these tissues.

The strongly tissue-specific expression patterns of 75 *DobHLH* genes were observed. A group of *bHLH* genes in *C. tinctorius* was found to exhibit high transcripts in petals [48]. During the flower development of jujube, *ZjbHLH62* and *ZjbHLH53* genes were stably expressed while the expression levels of *ZjbHLH4*, *12*, *23*, *78*, and *87* genes decreased [25]. There were seven *bHLH* genes with high expression levels in the petals of wheat [49]. *DobHLH96* was specifically and abundantly expressed in the flower bud, which suggests its potential role in the development of the flower bud. Moreover, the expression levels of *DobHLH74* in flower organs were much higher than that in the vegetative tissues. Its homolog in Arabidopsis is AtbHLH31 (At1G59640, BIGPETAL, BPE), which participated in the control of petal size and under the regulation of APETALA3, PISTILLATA, APETALA1, PISTILLATA3, and AGAMOUS [55].

Some *bHLH* genes could be highly expressed in roots such as *DobHLH12* and *DobHLH93*. In other species, some *bHLH* genes were also found to show abundant expression levels in roots such as five *bHLH* genes in wheat [49] and as many as 78 *bHLH* genes in maize [28]. These results suggest bHLH could play a conserved role in root development. Moreover, we found that some *bHLH* genes were abundant in the green root tip, implying that it could function to maintain root apical meristem. Some *DobHLH* genes had abundant transcripts in leaf. For example, *DobHLH54* had three cis-elements in its promoter region and may play a role in the process of photosynthesis.

MYC2 is an important regulator in the JA signaling pathway. In the present study, four putative MYC2 homologs were identified. A homolog of MYC2 (DobHLH33) in *D. officinale* was reported [56], which could be induced by MeJA and was involved in the biosynthesis of terpenoid indole alkaloids through the key enzyme genes in the MVA pathway. Moreover, four putative MYC2 homologs were expressed in roots as well as other tissues, suggesting their various functions in multiple tissues. Notably, DobHLH30 had the highest expression in the column (231.28). *MYC2* (*DobHLH33*) also had abundant expression in flower organs, which could have implications for functioning in flower development.

Usually, bHLH proteins function through the formation of homodimers or heterodimers to co-bind with their target genes to regulate their expressions. Many studies have reported the interactions of bHLH proteins to regulate the expressions of targeted genes [25,31,42,48,57]. Our current study found that many DobHLH proteins did have potential interactions via model prediction. In addition, we verified protein interactions between several DobHLH proteins such as DobHLH 25 and 26, and DobHLH17 and 60. We verified protein interactions between limited DobHLH proteins in this study. Many DobHLH proteins may function together to regulate plant growth and development, responses to environmental changes, and secondary metabolite processes in *D. officinale*. Additionally, we noted that DobHLH17 and DobHLH60 are homologous to UNE12 and PYE in Arabidopsis, and to ZmbHLH114 and ZmbHLH163 in maize, whose interactions have been verified in previous studies [28,57]. It is likely that protein interactions between given bHLHs proteins might be, at least partially, conserved in plants.

## 4. Materials and Methods 

### 4.1. Plant Materials

The seedling plants of *D. officinale* in vitro (provided by Prof. Shibao Zhang’s group) were well grown in the tissue culture room of the Kunming Institute of Botany, Chinese Academy of Sciences (Kunming, Yunnan, China). The medium was Murashige and Skoog (MS) medium with 30 g/L sucrose, and the growth was 25 °C with a constant photoperiod (12 h light/12 h dark) with an illumination of 40 µmol quanta/(m^2^ s). The fresh and healthy leaves, stem, and roots were harvested, immediately frozen in liquid nitrogen and stored at −80 °C for RNA extraction. 

Three individual *D. offcinale* plants of six months old in vitro were used for each treatment. MeJA (Coolaber, China) and ABA (Sangon, China) solutions were prepared with ethanol and water. The treated plants were sprayed with 100 µM MeJA or 100 µM ABA for three or six hours while the control plants were sprayed with ethanol and water. The leaves were harvested and quickly frozen at −80 °C for further RNA extraction.

### 4.2. Identification of the bHLH TF Family in Dendrobium officinale

Although two research groups reported the genome of *D. officinale* (*D. catenatum*), respectively [2,39], at present only one genome (PRJNA262478) is available. The protein data of *D. officinale* were downloaded from NCBI. HMMER was used to identify the potential bHLH proteins with the bHLH domain (PF00010) [58]. The presence of the bHLH domain was manually confirmed by CDD and SMART. After repeated or uncomplete proteins were discarded, the remaining proteins with the bHLH domain were considered as the bHLH candidate members in *D. officinale*. The isoelectric point and molecular weight of candidate DobHLH proteins were analyzed with the ProtParam tool [59]. The subcellular localization of DobHLH proteins was predicted by WOLF PSORT (https://www.genscript.com/psort.html).

### 4.3. Phylogenetic Analysis of DobHLH Proteins

The AtbHLH proteins of Arabidopsis were downloaded from TAIR. The proteins of rice bHLHs were also downloaded [40]. The sequences of bHLH proteins from these species were aligned with ClustalW software [60], and then an un-rooted tree was obtained with MEGA 7.0 software using a neighbor-joining method with a bootstrap of 1000 replicates [61].

### 4.4. Gene Structure and Conserved Motif Analysis of DobHLHs

The gene structures of *DobHLH* genes were analyzed by comparing cDNA sequences with the correspondent genomic DNA sequences with GSDS [62]. MEME was used to search 10 conserved motifs in DobHLH proteins [63] and the InterPro database was used to further analyze those motifs [64].

### 4.5. Expression Patterns of DobHLH Genes in Different Tissues

The previously reported RNA-Seq data of eight *D. officinale* tissues were used to analyze the expression patterns of *DobHLH* genes. First, the raw sequencing reads (PRJNA348403) were downloaded from the NCBI [65]. After low-quality reads and adapter sequences were filtered, all the clean reads were mapped to the reference genome of *D. officinale*, resulting in unique reads using HISAT2 software [66]. Then, transcript abundances were estimated with the FPKM method using Stringtie software [66]. The heatmap of DobHLH genes was obtained by the R package pheatmap, v1.0.10 (https://rdrr.io/cran/pheatmap/).

### 4.6. Predicted Protein–Protein Interaction of DobHLH Members

The potential protein–protein interactions were predicted with the online STRING server (https://string-db.org). The protein sequences of 98 DobHLHs were uploaded into the server and *Arabidopsis thaliana* was chosen as the comparative organism. After the BLAST analysis was finished, the interaction internet of DobHLHs was generated with genes showing the highest bitscore.

### 4.7. Cis-Element Analysis in Promoter Regions of DobHLH Genes

The 2000 bp sequences at the upstream of the transcription start site for each *DobHLH* gene were downloaded from the NCBI. Cis-regulatory elements were analyzed with PlantCARE.

### 4.8. RNA Extraction and cDNA Preparation

The total RNA was extracted with the RNAprep Pure Plant Plus Kit (Cat DP441, Tiangen, China) according to the manufacturer’s protocol. The concentration and quality of RNA samples were detected on a Nanodrop 2000 (Thermo Scientific, Waltham, MA, USA). cDNA was prepared with the one-step cDNA Synthesis Kit (Cat AT311, Transgen, China), which was used for DobHLH gene amplification in yeast two hybrid assays. A sample of 200 ng of total RNA in a reaction system of 20 μL was converted to cDNA with TransScript All-in-One First-Strand cDNA Synthesis SuperMix for qPCR (Cat AT341, Transgen, China), which was used for qRT-PCR assays.

### 4.9. Yeast Two Hybrid Assays

The CDS sequences of 11 *DobHLH* genes were amplified with PCR using cDNA prepared as templates. The primers used are shown in Appendix A. All the sequences were confirmed to be correct by sequencing. Then, the sequences of *DobHLH25*, *30*, *32*, *55*, *56*, and *60* genes were cloned into the pDest22 construct (Invitrogen) with the GAL4 activation domain (AD) while the sequences of *DobHLH17*, *25*, *26*, *36*, *48*, and *70* genes were cloned into pDest32 (Invitrogen) with the GAL4 DNA-binding domain (BD) through recombination reaction. Yeast two hybrid (Y2H) was performed as previously described [67].

### 4.10. qRT-PCR Assays

One μL of prepared cDNA mixture was used for qRT-PCR in a 20 μL volume with SuperReal PreMix Plus SYBR Green Kit (TIANGEN Biotech, China), according to the manufacturer’s instructions. The reaction was carried out on an Applied Biosystems QuantStudio 6 Flex Real-Time PCR System (ThemoFisher Scientific, USA) and the cycling conditions was as follows: first at 95 °C for 15 min, then 40 cycles of at 95 °C for 10 s, at 60 °C for 20 s, and finally at 72 °C for 32 s. The melt curves were performed at 95 °C for 15 s, at 60 °C for 1 min, and at 95 °C for 15 s. The 2^-ΔΔCT^ method was used to calculate Log_2_fold change with *DoActin* gene as an internal reference [13,68]. CT values stand for the average of cycle times from three technical replicates. The primer sequences used for qRT-PCR are listed in Appendix A.

## 5. Conclusions

This research presents a genome-wide identification and characterization of the bHLH TF family in *D. officinale*. The phylogenetic analysis found that 98 DobHLH members were classified into 18 clades, which showed conserved gene structures and motifs. The characteristics of the DobHLH domains for their function were predicted. Furthermore, the transcript data of *DobHLH* genes revealed their universal or specific expression profiles, which may contribute to understanding their function in *D. officinale*. Therefore, the present study provides additional insights into the bHLH TF family and will be helpful for the further investigation of DobHLH functions in *D. officinale*.

## Figures and Tables

**Figure 1 plants-09-01044-f001:**
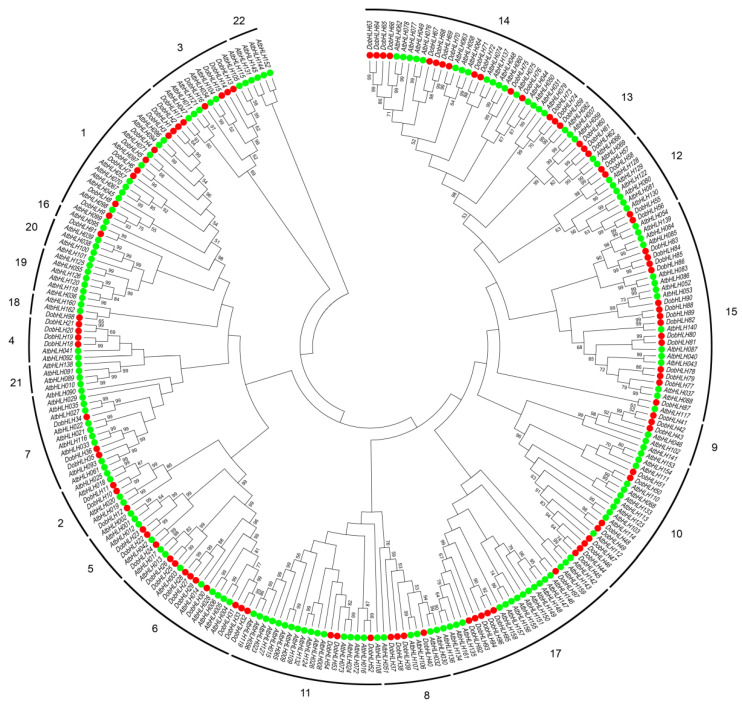
Phylogenetic analysis of bHLH proteins from *D. officinale* and Arabidopsis. A total of 98 DobHLH proteins and 162 AtbHLH from Arabidopsis (Arabidopsis thaliana) were used to construct the unrooted neighbor-joining (NJ) tree with a bootstrap of 1000 replicates. The bHLH proteins are grouped into 22 clades, and marked in different colors: *AtbHLH* genes are in green while *DobHLH* genes are in red.

**Figure 2 plants-09-01044-f002:**
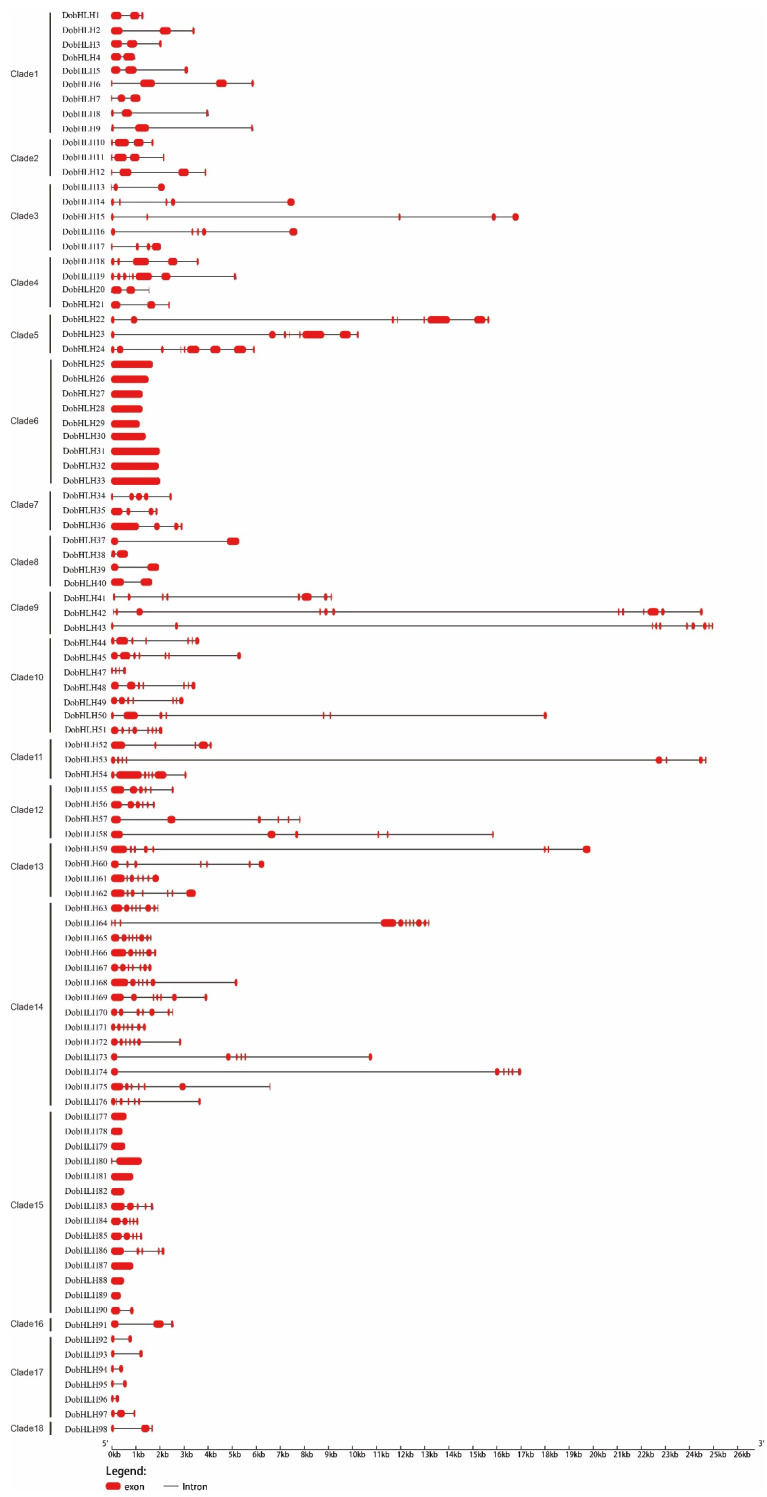
The gene structures of *DobHLH* genes in *D. officinale*. The exon-intron structures of 98 *DobHLH* genes were obtained with GSDS 2.0. The red rectangles and black lines indicate exons and introns, respectively, according to their lengths.

**Figure 3 plants-09-01044-f003:**
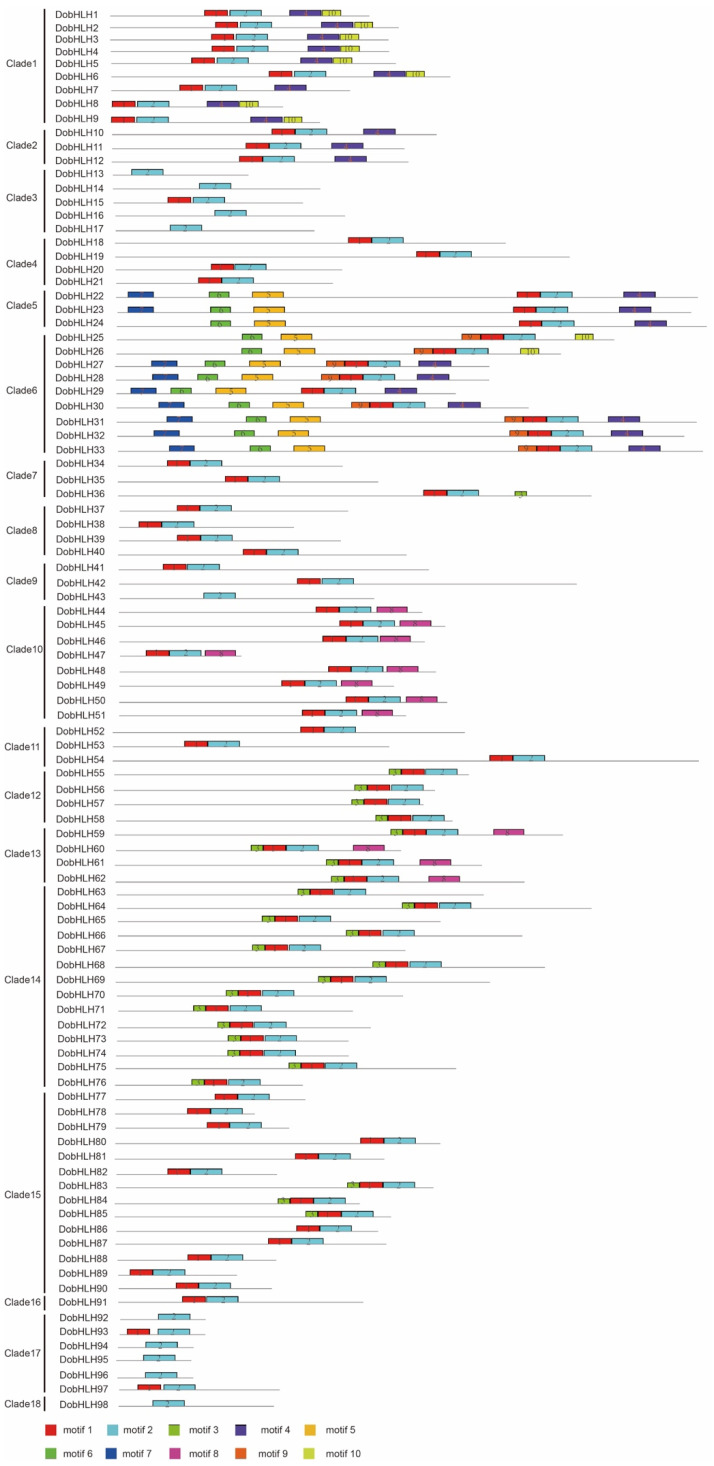
The conserved motifs of DobHLH proteins predicted by MEME. The colored boxes with numbers represent 10 conserved motif and the grey lines indicate non-conserved lines.

**Figure 4 plants-09-01044-f004:**
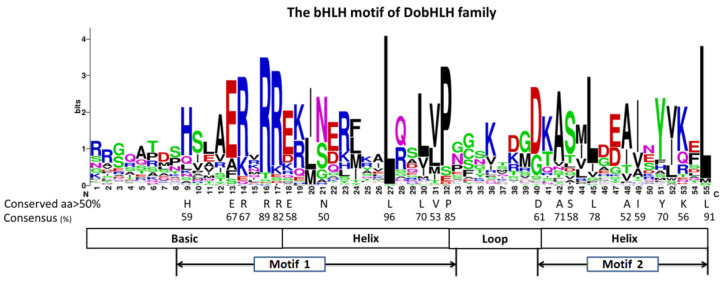
Analysis of the conserved amino acids in all DobHLH domains. The overall height of each amino acid represents the sequence conservation at that position. The conserved AAs and the consensus of more than 50% among 98 DobHLH domains are indicated in black letters.

**Figure 5 plants-09-01044-f005:**
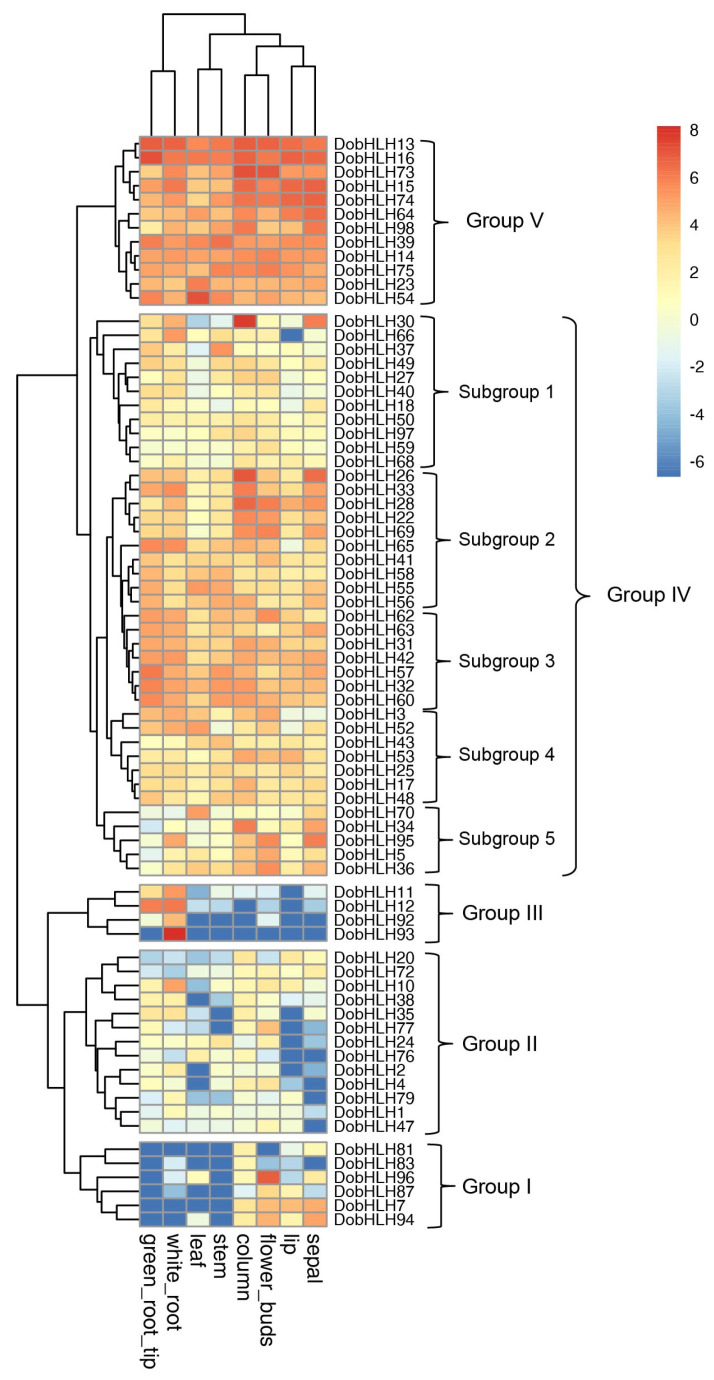
Expression patterns of 75 *DobHLH* genes in eight tissues. The expression levels of 75 *DobHLH* genes were from the RNA-Seq data. The eight samples included the column, flower buds, lip, sepal, leaf, stem, white root and green root tip. The color scale represents the values of Log_2_(FPKM + 0.01).

**Figure 6 plants-09-01044-f006:**
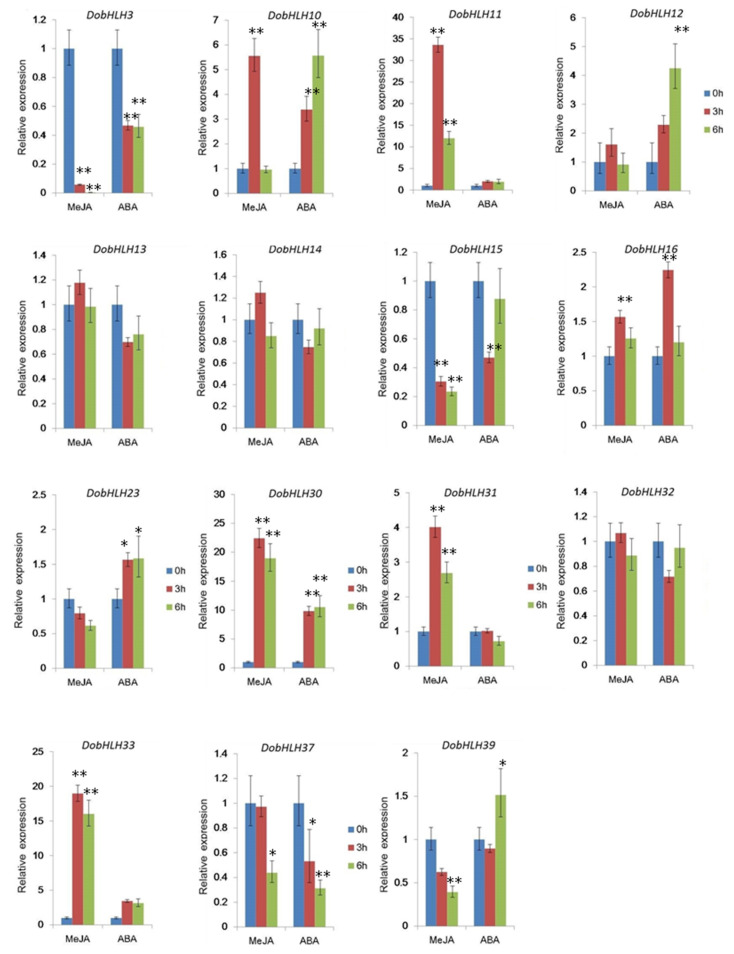
Expression patterns of 15 stress-related *DobHLH* genes under exogenous MeJA and ABA treatments by the qRT-PCR technique. After exogenous MeJA and ABA treatments in leaf tissues, the relative expression levels at 0, 3, and 6 h were tested with three biological duplicates. The relative expression levels at the beginning (0 h) were applied as the control. The bars denote the standard deviation. The star indicates the significance difference (* at *p* < 0.05, ** at *p* < 0.01) with the Student’s *t*-test.

**Figure 7 plants-09-01044-f007:**
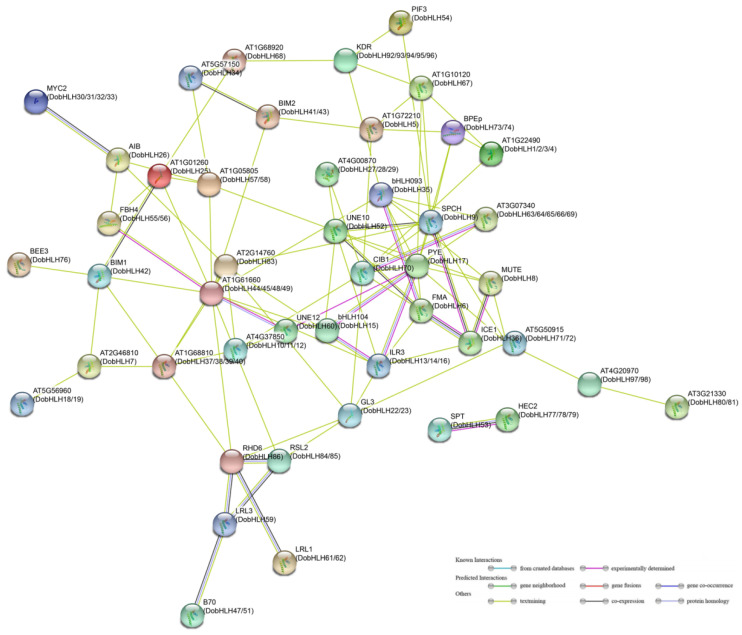
The predicted network of protein–protein interactions between DobHLHs by STRING. The different colors represent different types of interactions. Arabidopsis bHLH names are marked while their homologs in *D. officinale* are in parentheses.

**Figure 8 plants-09-01044-f008:**
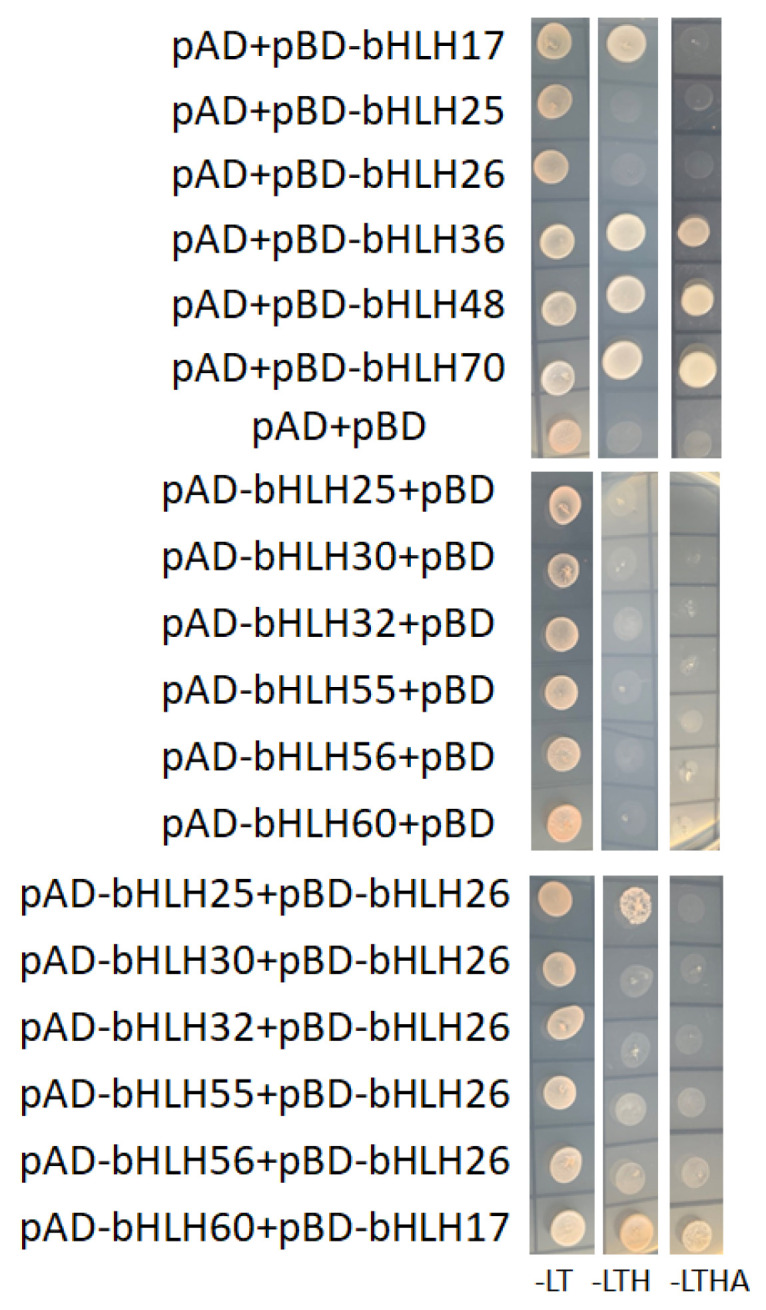
The protein–protein interactions between DobHLHs detected by Y2H. The sequences of DobHLH25, 30, 32, 55, 56, and 60 genes were ligated to the GAL4 activation domain (AD) while the sequences of DobHLH17, 26, 36, 48, and 70 genes were fused to the GAL4 DNA-binding domain (BD). SD-LT, SD-LTH, and SD-LTHA represent the SD-LeuTrp, SD-Leu-Trp-His, and SD-Leu-Trp-His-Ade medium, respectively.

**Table 1 plants-09-01044-t001:** The list of 98 *DobHLH* genes in *D. officinale.*

Gene Name	Gene ID	NCBI References	Clade	Gene Name	Gene ID	NCBI References	Clade
DobHLH1	LOC110116721	XP_020706080.1	1	DobHLH50	LOC110116300	XP_020705489.1	10
DobHLH2	LOC110115756	XP_020704772.1	1	DobHLH51	LOC114579352	XP_028549432.1	10
DobHLH3	LOC110107102	XP_020692919.1	1	DobHLH52	LOC110095032	XP_020676060.1	11
DobHLH4	LOC110096422	XP_020678032.1	1	DobHLH53	LOC110113329	XP_020701532.1	11
DobHLH5	LOC110116474	XP_020705700.1	1	DobHLH54	LOC110113204	XP_020701328.1	11
DobHLH6	LOC110098508	XP_020681021.1	1	DobHLH55	LOC110114625	XP_020703217.1	12
DobHLH7	LOC110111619	XP_020699225.1	1	DobHLH56	LOC110106166	XP_020691604.1	12
DobHLH8	LOC110112576	XP_020700508.1	1	DobHLH57	LOC110112285	XP_020700113.1	12
DobHLH9	LOC110108487	XP_020694816.2	1	DobHLH58	LOC110108563	XP_020694908.1	12
DobHLH10	LOC110110298	XP_020697356.1	2	DobHLH59	LOC110101829	XP_020685553.1	13
DobHLH11	LOC110093619	XP_020674221.1	2	DobHLH60	LOC110095526	XP_020676768.1	13
DobHLH12	LOC110115484	XP_020704390.1	2	DobHLH61	LOC110111433	XP_028555769.1	13
DobHLH13	LOC110093741	XP_020674407.1	3	DobHLH62	LOC110093428	XP_020673967.1	13
DobHLH14	LOC110102219	XP_020686105.1	3	DobHLH63	LOC110104725	XP_020689612.1	14
DobHLH15	LOC110098909	XP_020681520.1	3	DobHLH64	LOC110102342	XP_028552167.1	14
DobHLH16	LOC110107046	XP_028551992.1	3	DobHLH65	LOC110107722	XP_020693735.1	14
DobHLH17	LOC110096269	XP_020677790.1	3	DobHLH66	LOC110100649	XP_020683910.1	14
DobHLH18	LOC110116147	XP_020705280.1	4	DobHLH67	LOC110114258	XP_028550306.1	14
DobHLH19	LOC110094813	XP_028551573.1	4	DobHLH68	LOC110114447	XP_028556766.1	14
DobHLH20	LOC110107826	XP_020693884.2	4	DobHLH69	LOC110109277	XP_028547820.1	14
DobHLH21	LOC110094726	XP_020675682.1	4	DobHLH70	LOC110096494	XP_020678137.1	14
DobHLH22	LOC110114654	XP_020703261.1	5	DobHLH71	LOC110111863	XP_028551171.1	14
DobHLH23	LOC110111891	XP_020699606.1	5	DobHLH72	LOC110107433	XP_028552737.1	14
DobHLH24	LOC110097687	XP_020679864.1	5	DobHLH73	LOC110114740	XP_020703378.2	14
DobHLH25	LOC110103241	XP_020687528.1	6	DobHLH74	LOC110095482	XP_020676694.1	14
DobHLH26	LOC110094094	XP_020674906.1	6	DobHLH75	LOC110099461	XP_020682274.1	14
DobHLH27	LOC110116682	XP_020706025.1	6	DobHLH76	LOC110099214	XP_028548138.1	14
DobHLH28	LOC110099101	XP_020681804.1	6	DobHLH77	LOC110098199	XP_020680601.1	15
DobHLH29	LOC110113808	XP_020702166.2	6	DobHLH78	LOC110098270	XP_020680696.1	15
DobHLH30	LOC110094435	XP_020675326.1	6	DobHLH79	LOC110094287	XP_020675140.1	15
DobHLH31	LOC110116479	XP_020705705.1	6	DobHLH80	LOC110113891	XP_020702260.1	15
DobHLH32	LOC110114462	XP_020703009.1	6	DobHLH81	LOC110112441	XP_020700324.1	15
DobHLH33	LOC110092865	XP_020673218.1	6	DobHLH82	LOC110101026	XP_020684454.1	15
DobHLH34	LOC110109085	XP_020695652.1	7	DobHLH83	LOC110111081	XP_020698439.1	15
DobHLH35	LOC110096203	XP_020677673.1	7	DobHLH84	LOC110104845	XP_020689772.1	15
DobHLH36	LOC110112097	XP_020699850.1	7	DobHLH85	LOC110095998	XP_020677402.1	15
DobHLH37	LOC110107930	XP_020694038.1	8	DobHLH86	LOC110110399	XP_020697513.2	15
DobHLH38	LOC110115493	XP_020704404.1	8	DobHLH87	LOC110109507	XP_028547425.1	15
DobHLH39	LOC110094861	XP_020675853.1	8	DobHLH88	LOC114578594	XP_028547764.1	15
DobHLH40	LOC110112072	XP_020699818.1	8	DobHLH89	LOC110105526	XP_020690722.2	15
DobHLH41	LOC110114469	XP_028556744.1	9	DobHLH90	LOC110100961	XP_020684346.1	15
DobHLH42	LOC110107963	XP_020694091.1	9	DobHLH91	LOC110112336	XP_020700191.1	16
DobHLH43	LOC110106259	XP_028550041.1	9	DobHLH92	LOC110110400	XP_020697514.1	17
DobHLH44	LOC110107318	XP_028547365.1	10	DobHLH93	LOC110112399	XP_028555091.1	17
DobHLH45	LOC110103817	XP_028551326.1	10	DobHLH94	LOC110108630	XP_020695011.1	17
DobHLH46	LOC110114754	XP_020703396.1	10	DobHLH95	LOC110110529	XP_020697710.1	17
DobHLH47	LOC110107832	XP_020693899.1	10	DobHLH96	LOC110108826	XP_020695307.1	17
DobHLH48	LOC110107031	XP_020692824.1	10	DobHLH97	LOC110105593	XP_020690812.1	17
DobHLH49	LOC110098511	XP_020681024.1	10	DobHLH98	LOC110100228	XP_020683310.1	18

**Table 2 plants-09-01044-t002:** The predicted types of DNA-binding based on DobHLH domain sequences.

Predicted DNA-Binding Types Based on the bHLH Domain	Number of DobHLHs	The Ratio
DNA binding		
E-box	65	62.25%
G-box	61	62.24%
Non-G-box	4	4.08%
Non-E-box	11	11.22%
Total	76	77.55%
Non-DNA binding	22	22.45%

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
