# Peer review of "Genomic Characterization and Expression Analysis of Basic Helix-Loop-Helix (bHLH) Family Genes in Traditional Chinese Herb Dendrobium officinale"

_plants, 2020, doi:10.3390/plants9081044_

Round 1
Reviewer 1 Report
The manuscript describes bHLH genes from D. officinale. The aims and approach are simple, using publically available genomic and transcriptomic data to identify and characterise bHLH genes. The bHLH genes were categorised according to structure, motifs and tissue expression. Some of the predicted protein-protein interactions were confirmed by yeast two hybrid assays.
The manuscript is descriptive, but will be of use to future research in this species.
Some minor comments to improve understanding of the manuscript.
Figure 2: The genes could be divided according to clade (as in Fig. 3), to illustrate the differences between the clades as mentioned in the text.
Line 116: correct the higher PI value – “isoelectric point of 4.41-1078”
Lines 270-276: Does “limitedly expressed” mean low expression? Or expression in a small number of different tissues?
There are many English language and grammatical mistakes in the manuscript. Some are mentioned below, but there are many others. Thorough English language editing and proofing should be done.
Line 48: in the phrase “...makes slow growth and instable quality” does instable quality refer to the entire plant, or to the quality of the secondary metabolites?
Lines 57-58: the meaning of “....following that, he alkaloid biosynthesis in D. officinale is modification....” is not clear, please rephrase.
Lines 59-60: please correct the sentence “However, the transcription regulation of them biosynthesis has been largely unknown.”
Line 117-118: correct language “….were localization in the nuclear and a few were in the cytoplasmic or mitochondrial.”
Line 238: change “The most cis-element was G-box, which…..” to “The most frequent cis-element was G-box, which….” or similar.
Line 245: Change “…..can respond to defense and stress, wound,…..” to “…..can respond to defense and stress, wounding,…..”
Line 258: Change “There were 23 DobHLH genes that were hardly detected in none of the eight tissues….” To “Expression of 23 DobHLH genes was not detected in any of the eight tissues….”. If is is assumed that FPKM values below a certain threshold are considered to be not expressed, then that should be stated.
Line 279 and elsewhere: Change “…hardly or lowly expressed…” to “…not expressed or lowly expressed…”
Line 373-374: Change “…members from clade 7 and 11 in D. officinale were lowered than those in…” to “…members from clade 7 and 11 in D. officinale were lower than those in…”
Reviewer 2 Report
In this work, Yue Wang and Aizhong Liu identified 98 putative genes of bHLH transcription factors from Dendrobium officinale genome in order to characterize their chromosome positions, functional features for their genes, proteins and promoters, their tissue-specific expression as well as to identify protein-protein interactions between DobHLH. It should be noted that DobHLH transcription factors are involved in plant development as well as potential in regulation of biosynthesis of secondary metabolites and are characterized for a few plants, but the molecular mechanisms of their participation in these cellular processes of different plant species require clarification.
Initially, researchers searched for candidate genes that encode DobHLH transcription factors in the Dendrobium officinale genome. Further phylogenetic and feature domain analysis showed that (i) the typical highly conserved domains were present in majority of DobHLH proteins; (ii) the most of DobHLH proteins have putative nuclear localization signals and were in the cytoplasmic or mitochondrial; and (iii) conservative amino acid and DNA binding of DobHLH proteins is predicted, which is key to their function. Further, the researchers conducted a search for cis-acting elements in DobHLH gene promoters using the PLACE website in order to identify cis-acting elements associated with the hormone response in plants. Analysis of the transcriptome data available in the databases showed that gene expression from eight tissues showed that some DobHLH genes were expressed everywhere, while some DobHLH genes were expressed in specific tissues, indicating their possible functions in D. officinale.
Several experiments were performed to establish of the protein-protein interactions between DobHLH.
The topic of this work is interesting; however, there are some comments to manuscript:
Major comments:
- Most of the results were obtained due to in silico analysis of coronarium genomic and transcriptome data.
- However, the manuscript doesn’t contain a detailed description of the results of predicting the localization of DobHLH proteins in various plant cell compartments (as is done, for example, in Lei Zhang et al., Comparative Analysis and Expression Patterns of the PLP_deC Genes in Dendrobium officinale. Int. J. Mol. Sci. 2020, 21, 54; doi:10.3390/ijms21010054.
- the authors describe the results of the tissue-specific expression profiles of DobHLH genes only on the basis of the RNA-seq data available in the NCBI resource - PRJNA348403, it was extremely important to include data on experimental confirmation of tissue specificity by qRT-PCR.
- When the authors describe the results of the search and analysis of cis-elements in the promoter regions of DobHLH genes that are responsible for responses to MeJA, abscisic acid (ABA) and ethylene as well as in response to other stimuli, readers may find it interesting whether there are dependencies between the presence of regulatory cis-elements in DobHLH promoter regions and their functional role in the regulation of DobHLH gene transcription during treatment with different stimuli? In our opinion, expression patterns of DobHLH genes in D. officinale under treatments determined by qRT-PCR experiment are necessary.
- About the results regarding the results of matching the predicted network of protein-protein interactions between DobHLHs and experimental data on protein-protein interactions between DobHLHs. The authors indicate “it was predicted that DobHLH17 can interact with several DobHLH proteins, including DobHLH 6, 8, 9, 13, 14, 15, 16 and 60.” However, for experimental verification, DobHLH gene sequences 17, 25, 26, 30, 32, 32, 36, 48, 55, 56, 60, and 70 were used. A more precise argumentation of the DobHLH gene sequences selected for this experiment is needed.
- The Discussion section should be supplemented by analysis of experimental data on protein-protein interactions between DobHLH.
Minor comments:
- Lines 458-463 - Section 4.1. Plant material. “The seedlings plants of D. officinale in vitro were well grown in the tissue culture room of Kunming Institute of Botany, Chinese Academy of Sciences (Kunming, Yunnan, China). The medium was MS medium with 30 g/L sucrose and the growth was 25 °C with a constant photoperiod (12 h light/12 h dark).” – It must specify the light conditions (illumination in μmol quanta/(m2 s)) of the in the tissue culture room (growth chamber).
- Lines 462-463 and 501-505 - Section 4.1. Plant material and Section 4.8 RNA extraction and cDNA preparation, it is not clear what experiments used RNA and cDNA?
Reviewer 3 Report
- Consistency of concept. L175 to L193 dived bHLH into motif1 and motif
- The concept of motif1 and motif2 should be applied to the subsequent sections regarding bHLH, such as section 2.4 (Fig. 4) L198 to 199, L218 to L232. 2. Consistency of concept. Clade, subfamily, and groups are used to describe the organization of the DobHLH genes in different sections, particularly clade and subfamily. Are they meaning the same thing, or how are they related? If not, define them.
- Fig. 3 needs a color code for annotating the domains instead use of the numbers embodied in bars. The fonts are too small to see clearly.
- In Table 3, the number of DobHLHs and ratios (ration is a typo) are missing for E-box. BTW, what about the combinations of E-box and G-box?
- Section 2.4 identified the E-box and G-box binding bHLH domains, and 2.5 identified the cis elements including the G-box. One question would how many DobHLH genes have G-box in their promoter and meanwhile encode G-box-binding bHLH domain in their proteins? If so, these genes may under autoregulation.
- Need reference in L55-L56, L349 - L350, L396, L465 - L466, L485-L486. Need to indicate data in L404-L405.
- Rephase: L180 to L182,
- Delete: ", respectively" in L143; " and there were 3-35 members in each clade" in L146-L147; " and the less clades could be related to the less number of DobHLH members" in L367-L368; " Other motifs were focused in some specific clades." In L386.
- Many grammar problems, which can be fixed by "Gramarly" software.
Round 2
Reviewer 2 Report
After reviewing the resubmitted manuscript and the authors' responses to my comments, the following should be noted: (i) the authors satisfactorily responded to all the questions and comments I have raised; (ii) the comments and questions posed by me has been clarified in manuscript.
Author Response
Dear reviewer,
We are grateful for constructive suggestions and comments. According to your review report, we have checked English language in our manuscript throughout.
Best Regards,
Yue Wang
Reviewer 3 Report
Recognizing that you did tremendous work in characterizing the bHLH gene family in D.o., I found the major problem of this manuscript is the presentation and English writing. The English version was probably translated from a Chinese draft because it contains many Chinglish expressions. So I suggest you use an English language service to revise the whole paper. Following are some my major comments on presentation:
- The paragraphs from L46 to L100 need to re-organization to find the knowledge gap, i.e., why you selected bHLH gene family specifically for this research.
- What criteria were used from defining the clades in the phylogeny tree? OdbHLH87 is located in a major branch, but the branch was merged with others. How the clades were named? Why the clade numbers did not reflect the phylogeny relationship?
- The clades in Table 1, Figs. 1, 2, and 3 do not match. For example, DobHLH88 is in clade 15 in Figs. 1, 2, and 3 but was placed in clade 14 in Table 1.
- Paragraphs from L136 to L154 are boring. Revise the paragraph L136 to L147. Delete the paragraph L148 to L154 because D.o. is a dicot plant.
- Quality of Figs. 2 and 3 are very low, not suitable for publication. You need to remake the figs with high resolution for the drawings and the fonts. The yellow color in Fig. 2 for CDS (should be exons) is difficulty to read clearly. Increase the contrast.
- Move Table 2 to supplementary file because replicate the data presented in Fig. 4.
- Move paragraph L313-L318 to the section 2.7 and use it as the first paragraph of the section. Move Fig. 6 to supplementary file.
- For Fig. 7, statistics significance (* p , 0.05; ** P , 0.001, etc) should be indicated for fold change of 3h and 6h samples as compared to 0h samples.
